# The Flavoring Agent Ethyl Vanillin Induces Cellular Stress Responses in HK-2 Cells

**DOI:** 10.3390/toxics12070472

**Published:** 2024-06-29

**Authors:** Ashley J. Cox, Kathleen C. Brown, Monica A. Valentovic

**Affiliations:** Department of Biomedical Sciences, Toxicology Research Cluster, Marshall University Joan C. Edwards School of Medicine, Huntington, WV 25701, USA; saunders29@marshall.edu (A.J.C.); brown364@marshall.edu (K.C.B.)

**Keywords:** renal, flavoring agent, HK-2 cells, ethyl vanillin, cytotoxicity

## Abstract

Flavored e-cigarettes are a popular alternative to cigarette smoking; unfortunately, the extrapulmonary effects are not well-characterized. Human proximal tubule cells were cultured for 24 or 48 h with 0–1000 µM ethyl vanillin (ETH VAN) and cytotoxicity evaluated. Mitochondrial health was significantly diminished following 48 h of exposure, accompanied by significantly decreased spare capacity, coupling efficiency, and ATP synthase expression. ETH VAN at 24 h inhibited glycolysis. The endoplasmic reticulum (ER) stress marker C/EBP homologous protein (CHOP) was increased at 100 μM relative to 500–1000 μM. The downstream proapoptotic marker cleaved caspase-3 subsequently showed a decreasing trend in expression after 48 h of exposure. The autophagy biomarkers microtubule-associated proteins 1A/1B light chain 3 (LC3B-I and LC3B-II) were measured by Western blot. LC3B-II levels and the LC3B-II/LC3B-I ratio increased at 24 h, which suggested activation of autophagy. In contrast, by 48 h, the autophagy biomarker LC3B-II decreased, resulting in no change in the LC3B-II/LC3B-I ratio. Mitophagy biomarker PTEN-induced putative kinase 1 (PINK1) expression decreased after 48 h of exposure. The downstream marker Parkin was not significantly changed after 24 or 48 h. These findings indicate that the flavoring ETH VAN can induce energy pathway dysfunction and cellular stress responses in a renal model.

## 1. Introduction

Flavoring compounds such as ethyl vanillin have been declared “generally recognized as safe” (GRAS) by the Food and Drug Administration (FDA), but this classification is only for intended use as an ingested food ingredient. The GRAS label applies to the ingestion of flavoring agents such as ethyl vanillin and does not require pre-market review by the FDA. Nevertheless, the safety of inhaling vaporized flavoring ingredients remains relatively unknown. Electronic nicotine delivery systems (ENDSs) devices are widely popular, with national surveys reporting that nearly 1 in 5 preteens and adolescents have participated in “vaping” or “vaping nicotine” within the past 30 days [1,2]. ENDSs devices include products such as e-cigarettes, vape pens, vape pods, Puff Bars, and disposable or one-time-use vape products. These products contain ingredients including nicotine, flavors, and variable ratios of propylene glycol (PG) and vegetable glycerin (VG). E-liquids are heated and vaporized within ENDSs devices and then inhaled by the user. Flavorings have been shown to increase vaping and vaping initiation [3,4,5,6].

It has been previously reported that there are approximately 7800 e-liquid flavors on the market [7]. The flavoring ethyl vanillin (ETH VAN) is very popular and is an ingredient in products that are labeled as vanilla, dessert, baked goods, and tobacco [8,9]. ETH VAN has been correlated to cytotoxicity in murine neuronal stem cells (mNSCs), lung epithelial BEAS-2B cells [10,11], and liver HepG2 cells [12]. ETH VAN and its corresponding PG-acetal form can also activate transient receptor potential ankyrin 1 (TRPA1), which has been linked to respiratory diseases [13,14]. In addition, ETH VAN can inhibit CYP2A6 activity, which may play a role in promoting addictive behavior [15,16]. These results indicate the use of ETH VAN in e-liquid products warrants further investigation when determining its safety.

There are many studies that focus on the effects of flavorings on the respiratory tract, which receives the first exposure to inhaled e-liquids. However, effects beyond the lung are not well known. The kidney is the site of filtration of the blood and likely encounters e-liquid ingredients and their metabolites after they exit the respiratory tract. Changes in oxidative stress markers in the kidney, along with changes in the collecting duct, have been observed in rats exposed to nicotine-free, tobacco-flavored e-liquid [17]. Mice on a high-fat diet (HFD) exposed to nicotine-free vapor long-term have shown decreased mitochondrial OXPHOS and manganese superoxide (MnSOD) activity in the kidney. ENDS use has not been associated with kidney disease, but there have been clinical case reports of users presenting with renal failure, acute kidney injury (AKI), and renal infarction [18,19,20]. Therefore, this study focused on characterizing targets of renal cell damage mediated by ETH VAN. Apoptosis, lipid peroxidation, endoplasmic reticulum (ER) stress, and autophagy were measured after exposing HK-2 cells to ETH VAN. MTT assay and trypan blue exclusion were measured to quantitate mitochondrial status and cell viability. Both glycolytic and mitochondrial energy were measured using a Seahorse Analyzer.

## 2. Materials and Methods

### 2.1. Chemicals and Reagents

Ethyl vanillin, 3-ethoxy-4-hydroxybenzaldehyde (CAS Number 121-32-4, Beilstein Number 1073761), was obtained from Sigma-Aldrich (St. Louis, MO, USA, Item No. W246409-1KG-K, ethyl alcohol (EtOH), 200 proof, USP was the solvent used to prepare ethyl vanillin treatments, and was procured from Koptec (King of Prussia, PA, USA, Item No. V1001). 

### 2.2. Cell Culture and Treatment

HK-2 cells from American Type Culture Collection (ATCC, Manassas, VA, USA, CRL-2190) were cultured according to vendor recommendations, which specify keratinocyte-free media supplemented with 50 µg/mL bovine pituitary extract and 5 ng/mL recombinant epithelial growth factor sourced from Thermo Fisher Scientific (Gibco, Carlsbad, CA, USA, Item No. 17005-04). Penicillin–streptomycin (50 units/mL, 0.5% for both antibiotics) (Gibco, Grand Island, NY, USA, Item No. 15140-122) was added to each flask to inhibit bacterial growth. Cells were grown in T75 flasks (USA Scientific, CytoOne, Item No. CC7682-4875) with 6.0 × 10^6^ cells per flask. The HK-2 cells were treated with 0 (EtOH)-1000 µM ETH VAN for 24 or 48 h. The maximum percent ethanol was 1% of the total volume per well. Cells were collected at the end of the exposure for viability, mitochondrial function, and Western blot analysis. All studies were completed as 4 independent experiments with dissimilar cell passages.

### 2.3. Trypan Blue Exclusion

Cells were incubated for 24 or 48 h with 0–1000 µM ETH VAN and then harvested. An aliquot was added to an equal volume of 40% *w*/*v* trypan blue solution (Sigma Aldrich, Item No. T6146). After gentle mixing, the mixture (10 µL) was quantitated using a Countess II FL cell counter (Thermo Fisher Scientific) for total, living, and dead cell numbers.

### 2.4. MTT Assay

For the MTT assay, 3-(4,5-dimethylthiazol-2-yl)-2,5-diphenyltetrazolium bromide (MTT) was procured from Sigma Aldrich, St. Louis, MO, Item No. 1002120223. HK-2 cells (1 × 10^4^/well) were seeded in 96-well plates (Fisher Scientific, Item No. FB012931) and equilibrated for 48 h. The media was replaced and cells were incubated with 0–1000 µM ETH VAN. Next, MTT (5 mg/mL) was allowed to react with cells for 4 h in the absence of light at room temperature with slight oscillation. Finally, the wells were aspirated, 100 µL of DMSO (Fisher Scientific, Fair Lawn, NJ, Item No. D128-1) was added to solubilize the formazan, and the absorbances was measured at 570 nm. 

### 2.5. Western Blot Analysis

The cells (1.0 × 10^6^ cells/well) were seeded in 6-well plates and equilibrated for 2 days. The cells were treated, as described above, for 1 or 2 days. After treatment, cells were harvested using trypsin (Gibco, Grand Island, NY, Item No. 25200-072), rinsed with Krebs buffer, and lysed (Cell Signaling Technology, Danvers, MA, USA, Item No. 9803). Western blot analysis was completed with primary antibodies: 4-hydroxynonenal (4-HNE, 1:1000; EMD Millipore, Billerica, MA, USA, Item No. ABN249), autophagy-related 7 (ATG7, 1:1000; Abcam, Waltham, MA, USA, Item No. 133528), C/EPB homologous protein (CHOP, 1:1000; Cell Signaling Technology, Danvers, MA, Item No. 2895S), microtubule-associated protein 1A/1B light chain 3 (LC3B, 1:1000; Abcam, Waltham, MA, USA, Item No. 48394), mitochondrial oxidative phosphorylation complexes (OXPHOS, 1:1000; Abcam, Cambridge, MA, Item No. 110413), p62 (1:1000; Abcam, Item No. 56416), Parkin (1:500; Abcam, Item No. 77924), PINK1 (1:1000; Abcam, Waltham, MA, Item No. 300623; Novus Biologicals, Centennial, CO, BC100-494; Fisher Scientific, Invitrogen, Rockford, IL, PA1-16604), and caspase 3 (1:1000; Abcam, Cambridge, MA, Item No. 136812). Protein was measured using Bradford assay [21], and each sample (35 µg protein) was adjusted to a final volume of 20 µL with water (ddH2O) and combined with reducing sample buffer (RSB). Prior to loading, samples were subjected to boiling for 5 min, with the exception of the OXPHOS samples. Samples were run on 12.5% polyacrylamide gel and transferred to nitrocellulose membrane (Bio-Rad, Hercules, CA, USA, Item No. 1620097). The transfer was verified by a Memcode Reversible Stain Kit (Fisher Scientific, Pierce Biotechnology, Rockford, IL, Item No. PI-24580). Proteins were blocked for 1 h with 1% *w*/*v* bovine serum albumin (BSA) in TBST (10 mM Tris-HCl, 150 mM NaCl, 0.1% Tween-20; pH 8.0) or 3% milk *w*/*v* in TBST, for 4-HNE or Parkin, respectively. The remaining membranes were blocked for 1 h using 5% milk *w*/*v* in TBST. Membranes were incubated overnight with primary antibodies at 4 °C. Primary antibodies were washed from the membranes with TBST 4× for 5–10 min, followed by adding secondary antibody (goat anti-rabbit HRP, 1:2000; Cell Signaling Technology, Danvers, MA, Item No. 7074; goat anti-mouse HRP, 1:2000; Cell Signaling Technology, Danvers, MA, Item No. 7076S) for 1–1.5 h, based on the vendor’s guidelines. The secondary antibody was rinsed for 5–10 min thrice with TBST and once for 5–10 min in TBS (10 mM Tris-HCl, 150 mM NaCl, pH 8.0). The enhanced chemiluminescence (ECL) solution was a 2:1:1 mixture of H_2_O_2_ (30% *v*/*v*), coumaric acid (90 mM), and luminol (250 mM) in 1 M Tris, pH 8.5, which was added to visualize bands. Densitometry analysis was performed with a Bio-Rad ChemiDoc system (Image Lab version 6.0.1, Bio-Rad, Hercules, CA, Item No. 170-9690).

### 2.6. Seahorse XFp Analysis

Oxygen consumption rate (OCR) and extracellular acidification (ECAR) were probed using Seahorse technology using cell mito and glycolysis stress test kits (Agilent, Cedar Creek, TX, USA, Item No. 103010-100 and 103017-100), respectively. HK-2 cells (1.5 × 10^4^/well) were seeded in XFp miniplates (Agilent Technologies, Cedar Creek, TX, USA, Item No. 102984-100) and equilibrated for 2 days. The media was replaced, and cells were incubated for 24 h with 0, 100, or 1000 µM ETH VAN. The cell mito stress test assay wells were rinsed twice with assay media (Agilent Technologies, Santa Clara, CA, USA, 103575-100) containing 2 mM glutamine, 1 mM pyruvate, and 10 mM glucose (Agilent, Santa Clara, CA, Item Nos. 103579-100, 103578-100, and 103577-100). The glycolysis stress tests have 2 mM glutamine added to the assay media. All plates were equilibrated in 175 µL assay media for 1 h at 37 °C in an incubator lacking CO_2_. The cell mito stress test sequentially injects: oligomycin (1.5 µM/well), carbonyl cyanide-4-(trifluoromethoxy)phenylhydrazone (FCCP, 0.5 µM/well), and rotenone/antimycin-A (0.5 µM/well). The glycolysis stress test kit sequentially injects: glucose (10 mM), oligomycin (1.0 µM), and 2-deoxy-glucose (2-DG, 50 mM), respectively. Measurements of OCR and ECAR were taken every 3 min throughout the test assay. Prior to beginning the glycolysis stress test, the cells were equilibrated at 37 °C for 1 h in media lacking glucose and pyruvate. At the end of the tests, the wells were rinsed twice with Krebs and frozen (−80 °C). 

### 2.7. Cell Normalization

Seahorse results were normalized using CyQUANT Direct Cell Proliferation Assay (Invitrogen Thermo Fisher Scientific, Life Technologies, Eugene, OR, Item No. C35011). Miniplates were thawed to room temperature. A working solution (WS) of cell lysis buffer (200 µL), ddH2O (3790 µL), and CyQUANT GR dye (10 µL) was prepared in a dark room. A 200 µL aliquot of WS was added to each well and equilibrated for 5 min in the dark. An aliquot (175 µL) of each well was pipetted to a black-walled, clear-bottomed 96-well plate (Fisher Scientific, Greiner Bio-One, Item No. 655906). Fluorescence was measured at 480 and 520 nm (BioTek plate reader, Gen5 software, version 3.14.03).

### 2.8. Statistical Analysis

Data are expressed as mean ± SEM with ≥4 independent experiments. Differences between groups were calculated via a one-way ANOVA followed by a Tukey post-hoc test, with values of *p* < 0.05 considered significant (GraphPad Prism Software, version 10.1.0).

## 3. Results

### 3.1. ETH VAN Alters Mitochondrial Activity

ETH VAN significantly increased mitochondrial activity at 24 h, which then returned to levels similar to the vehicle control (Figure 1A). However, after 48 h exposure, MTT declined relative to the control group in the 250–1000 µM ETH VAN concentration range (Figure 1B). Trypan blue exclusion, a biomarker of loss of cell membrane integrity, showed no significant differences between groups at any time period (Figure 1C,D).

### 3.2. ETH VAN Alters Mitochondrial Energy Pathways

Mitochondrial function showed that a 24 h exposure to ETH VAN had no significant changes in mitochondrial parameters (Figure 2A–G). However, increasing trends were detected in proton leak, non-mitochondrial respiration, maximal FCCP stimulated OCR, and spare capacity (Figure 2C–F). A representative 24 h time–course profile is shown in Figure 2H. 

In contrast, at 48 h, ETH VAN impaired spare capacity and coupling efficiency (Figure 3F,G). Other parameters were not different (Figure 3A–D), but when compared to 24 h exposure, these parameters were altered. Both basal respiration and ATP production decreased more relative to controls at 48 h when compared to 24 h exposure. In addition, maximal respiration decreased at both 100 and 750 µM at 48 h, whereas 24 h exposure caused this parameter to increase in the 1000 µM group.

### 3.3. ETH VAN Decreases Expression of ATP Synthase

In order to visualize changes to mitochondrial energy pathways, Western blotting was performed on the oxidative phosphorylation (OXPHOS) complexes. Complex V expression, also known as ATP synthase, was significantly reduced after 24 and 48 h of exposure to ETH VAN (Figure 4A,B). ETH VAN did not alter expression of Complexes I, II, and IV at either time point. 

### 3.4. ETH VAN Decreases Glycolytic Function

Glycolytic function was probed at 24 h. Exposure to ETH VAN for 24 h significantly decreased glycolysis, glycolytic capacity, and glycolytic acidification (Figure 5A,B,D). Glycolytic reserve showed a trend (*p* > 0.05) for a decline, which was not statistically significant (Figure 5C). 

### 3.5. ETH VAN Alters Cleaved Caspase-3 and Induces Loss of CHOP Expression

A biomarker of endoplasmic reticulum (ER) stress, C/EBP homologous protein (CHOP), is an upstream regulator of apoptosis. Unfolded or damaged proteins can initiate the unfolded protein response (UPR). However, if damage cannot be mediated, CHOP may be induced in order to initiate apoptosis. Cleaved and uncleaved caspase-3 were also evaluated after exposure to ETH VAN. CHOP expression was not detectable following 24 h of exposure to ETH VAN (Figure 6B). CHOP was induced at 48 h, but this expression was lost at higher concentrations of ETH VAN (Figure 6A,B). Uncleaved caspase-3 was not altered at any time period. At 24 h, an increasing trend for cleaved caspase-3 was detected (Figure 6C), but was not statistically significant. In contrast, after 48 h, cleaved caspase-3 expression showed a decreasing trend (Figure 6D).

### 3.6. ETH VAN Alters Autophagy Markers

Microtubule-associated protein 1A/1B light chain 3 (LC3B) is present in two forms, identified as LC3B-I and LC3B-II. LC3B-I is a soluble, cytosolic form that will conjugate with phosphatidylethanolamine (PE) to form LC3B-II, which assists autophagosome membrane binding during autophagy. The LC3B-II/LC3B-I ratio evaluates whether autophagy is triggered within the cell. No differences were detected in LC3B-I protein levels at either time point between all groups (Figure 7A,D). ETH VAN at 750 µM increased LC3B-II when compared to 0 μM (Figure 7B). Interestingly, LC3B-II decreased significantly by 48 h (Figure 7E). LC3B-II/LC3B-I was significantly increased (Figure 7C). This increase in ratio was lost by 48 h, and a decreasing trend was, instead, observed (Figure 7F). Figure 7G shows a representative blot. 

Additional biomarkers of autophagy were evaluated following ETH VAN treatment. The autophagy markers p62 and ATG7 were also probed. ATG7 interacts with and promotes PE binding to LC3B-I; p62 is a marker for cellular waste to be broken down by autophagy. ATG7 and p62 were not different between groups at 24 h. An initial increasing trend for p62 was observed after 48 h and subsequently decreased (Figure 7H); however, these changes were not significant (*p* > 0.05).

### 3.7. ETH VAN Alters Mitophagy Pathway Markers

Mitophagy is an organelle-specific autophagy involving mitochondria. Based on the MTT and Seahorse results, markers of mitophagy were probed. PTEN-induced putative kinase 1 (PINK1) may accumulate when mitochondria are damaged, and will attract Parkin. PINK1 expression showed a trend to decline at 24 h (Figure 8A) and decreased at 48 h at 750 and 1000 μM (Figure 8C). Parkin was similar between all groups (Figure 8B,D).

### 3.8. ETH VAN Mediates Lipid Peroxidation-Induced Oxidative Stress

Oxidative stress generates the reactive product, 4-hydroxynonenal (4-HNE), during lipid peroxidation; 4-HNE can covalently bind to proteins and cause them to become impaired or non-functional. The 4-HNE was unchanged at 24 h exposure to ETH VAN (Figure 9A). In contrast, at 48 h, ETH VAN induced significant decreases of 4-HNE expression between the control group and all others except the 1000 µM group (Figure 9B).

## 4. Discussion

ETH VAN induced a wide variety of cellular changes in HK-2 cells. MTT assay showed an initial increase in activity by 24 h exposure, but by 48 h, mitochondrial health had declined. The initial increase in function was accompanied by slight increases in mitochondrial parameters during the 24 h cell mito stress test experiments as well. In contrast, a significant decrease in ATP synthase protein expression was apparent after 24 and 48 h exposure. This is not surprising given that the 48 h cell mito stress test showed significant decreases in spare capacity and coupling efficiency, as well as slight decreases in ATP production and maximal respiration. The structure of ETH VAN is not particularly lipophilic, and so it may take longer to penetrate than other compounds through the mitochondrial membrane to exert its effects. In addition, it has previously been shown that ETH VAN is not as deleterious to mitochondrial health compared to its PG- acetal form [22]. Interestingly, trypan blue exclusion cell viability testing showed no significant changes after ETH VAN exposure. This suggests that while ETH VAN did affect mitochondrial function, it was not enough to cause loss of cell membrane integrity and ultimately cell death.

The glycolytic function tests show that at 24 h, ETH VAN significantly decreased glycolysis, glycolytic capacity, and non-glycolytic acidification. This could be compensation due to the increased mitochondrial output at 24 h. However, it is also possible that ETH VAN directly interacts with enzymes and other components in the glycolysis pathway, causing decreased energy output. In terms of energy requirements, the kidney is highly dynamic. Along with the heart, it has one of the highest resting metabolic rates when compared to other organs [23]. The kidneys also receive 25% of cardiac output. Due to this, changes to renal metabolic output as we have shown in the present study, could be deleterious.

In the present study, the ER and apoptosis results may be mediated by ETH VAN anti-oxidant properties. Initially, at 24 h, CHOP expression was unchanged by ETH VAN. However, at 48 h, CHOP levels were different between 100 μM and a marked decline at higher ETH VAN concentrations. CHOP is an upstream regulator of apoptosis and can activate the apoptotic pathway if the ER is overburdened with misfolded proteins. Cleaved caspase-3 was not altered at 24 h exposure, though an increasing trend was observed. Similarly, after 48 h of exposure, cleaved caspase-3 expression had not changed significantly. It is plausible that the loss of CHOP expression also contributed to the loss of caspase-3 cleavage, therefore decreasing apoptotic pathway activity.

Autophagy pathway markers were changed by ETH VAN exposure. LC3B-II and the LC3B-II/LC3B-I ratio were elevated at 24 h. In contrast, at 48 h, we showed that LC3B-II expression decreased significantly, which was accompanied by a decrease in LC3B-II/LC3B-I. Although it was not statistically significant, the expression of the autophagy marker p62 also followed the same trends after 48 h of exposure. If autophagy is initially triggered by ETH VAN exposure, it may be lost after 48 h of exposure, according to our results. Mitophagy markers PINK1 and Parkin were unchanged when compared to controls after 24 h. However, after 48 h of exposure, PINK1 expression significantly decreased while Parkin showed no significant change. Since the results showed a possible loss of autophagic signaling in LC3B at 48 h, the subsequent mitophagy pathway signaling may also be lost after ETH VAN exposure. However, it is not possible to determine whether or not this loss is due to autophagic malfunction. It has been shown previously that Parkin can recruit p62 to accumulate on the mitochondria [24,25], but it is unknown if p62 is essential to active mitophagy [26,27]. In addition, autophagy and apoptosis processes in the cell are interconnected [28,29], and alterations in these pathways, as the present study has demonstrated, may cause changes in each other. Finally, when autophagy and mitophagy are functional in the kidney, they act in a protective manner during renal tissue generation; however, if mitophagy processes become dysfunctional, tissue regeneration is actually impeded [30,31]. It is therefore important to understand the effects of ETH VAN in order to assess its safety.

We also showed that ETH VAN exposure can augment the oxidative stress marker 4-HNE and cause significant decreases in expression. ETH VAN has been shown to protect the kidney in a diabetic nephropathy rodent model by reducing oxidative stress and apoptosis, as well as increase antioxidant activity in the plasma of mice [32,33]. Similarly, the related flavoring vanillin can also mediate renal damage from cisplatin, methotrexate, thioacetamide, and bromate, as well as increase the presence of antioxidant enzymes [34,35,36]. Therefore, it is possible that ETH VAN antioxidant activity decreased 4-HNE expression.

One limitation of this study was the direct addition of flavoring to the media of the HK-2 cells. This type of exposure does not entirely represent how exposure occurs in ENDS users. After inhaling e-vapor, it is likely that its ingredients will be modified and biotransformed as they travel throughout the body. Currently, it is not known which metabolites are most abundant after inhalation of ETH VAN. Oral administration in rats results in metabolites such as ethyl vanillic acid, ethyl vanillyl alcohol, and glycine conjugates [37]. It may be useful in the future to screen ETH VAN metabolites in order to evaluate the safety of ETH VAN inhalation. Furthermore, there are thousands of commercially available flavoring brands on the market. This makes it challenging to identify the concentration of ETH VAN during normal exposure when using these products. In addition, a portion of ENDS users create their own e-liquids in a “do-it-yourself” (DIY) manner. One recent study has calculated the projected human consumption of ETH VAN through inhalation of an e-liquid over the course of 2 days to be 44.8 mg, assuming the initial concentration of ETH VAN to be 6.6 mg/mL and 3.4 mL of e-liquid use [38]. Other studies that quantified ETH VAN concentrations in commercial products have shown a range from 0.2 to 19.07 mg/mL [9,39]. Tierney and colleagues [9] reported e-liquid commercial products had concentration ranges of 1.2–50 mM. These concentrations exceed the levels used in this manuscript, which implies that ETH VAN may have cellular effects at very low concentrations, especially on the kidney. Given the changing market and DIY community, more research is needed in order to assess potential targets and concentrations for flavoring agents, including ETH VAN.

## 5. Conclusions

In conclusion, our study has characterized acute ETH VAN exposure effects in renal HK-2 cells. Our work has demonstrated significant changes in mitochondrial and glycolytic pathways, with significant mitochondrial output shown after 48 h of exposure. Similarly, significant decreases in glycolytic output indicate these energy effects are not limited to the mitochondria. We have also shown potential activation and subsequent loss of autophagy and mitophagy processes; however, more studies are needed to verify these effects. Furthermore, the present study has shown that ETH VAN exposure can cause a loss of CHOP expression, which may also cause changes in expression of the downstream proapoptotic marker cleaved caspase-3. Finally, we have shown that ETH VAN exposure can ameliorate expression of 4-HNE generated by lipid peroxidation. Taken together, these results indicate that ETH VAN can induce cellular stress responses in a renal model. More studies will be needed in the future to characterize these effects and to assess the safety of using this flavoring in various products.

## Figures and Tables

**Figure 1 toxics-12-00472-f001:**
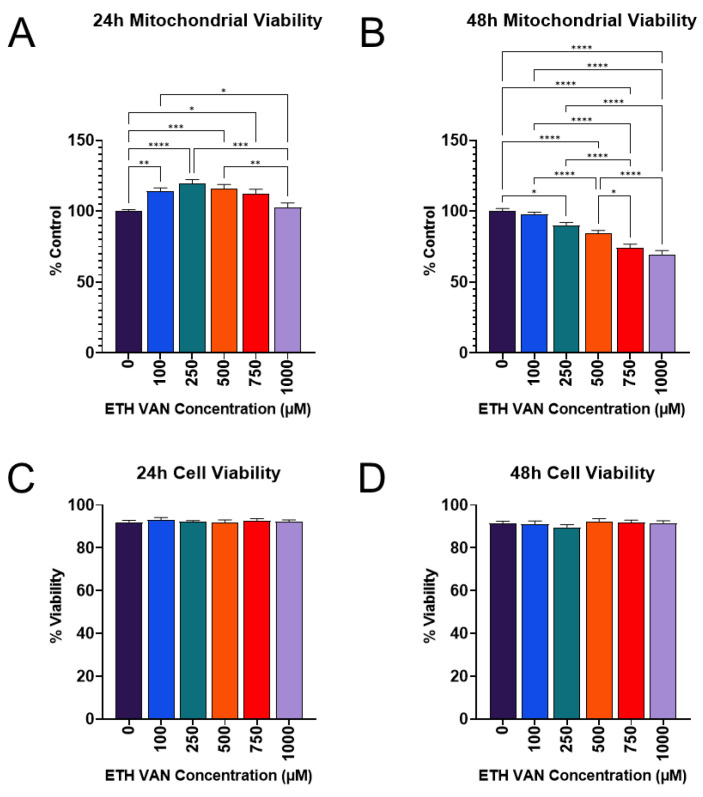
**Ethyl vanillin effects on mitochondrial and cell viability.** ETH VAN at 24 h increased mitochondrial health followed by decrease (**A**), while 48 h exposure significantly reduced mitochondrial health (**B**). Trypan blue exclusion was not different between all groups at 24 or 48 h (**C**,**D**). One-way ANOVA was used for statistical analysis, followed by a Tukey post-hoc test. Statistical differences denoted by an asterisk (* *p* < 0.05, ** *p* < 0.01, *** *p* < 0.001, **** *p* < 0.0001). Values represent mean ± SEM for a minimum of 4 independent experiments using different biological replicates.

**Figure 2 toxics-12-00472-f002:**
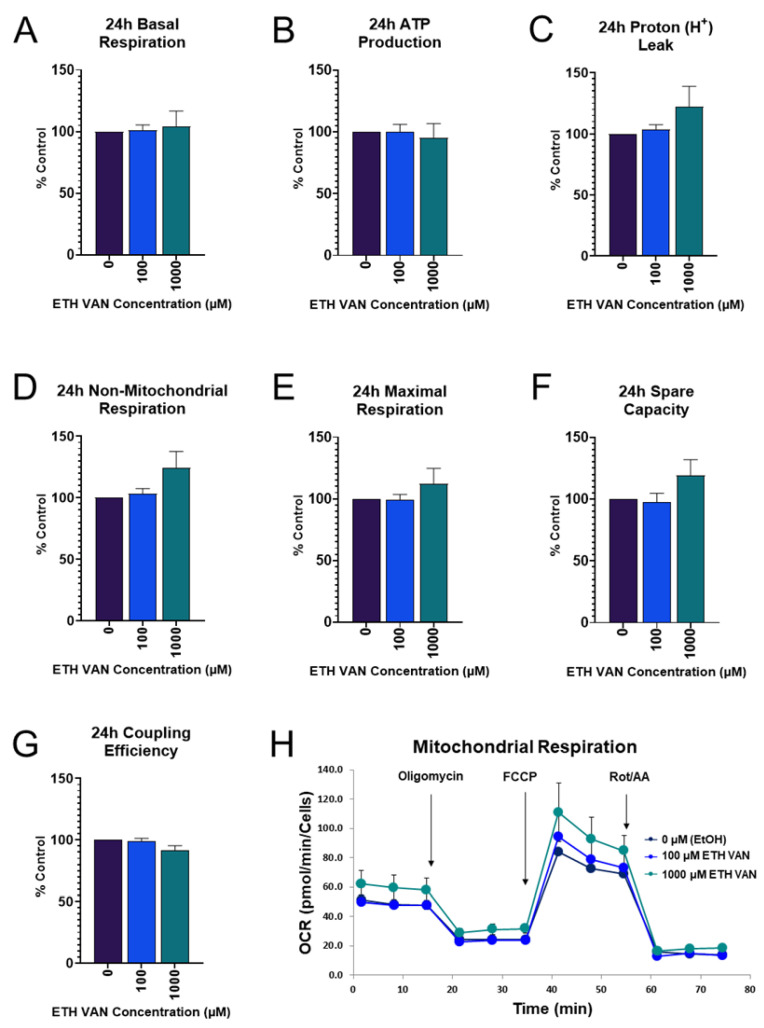
**Ethyl vanillin effects on mitochondrial function after 24 h.** No differences were detected at 24 h in mitochondrial parameters (**A**–**G**). Representative output for cell mito stress test at (**H**). Normalization was conducted with CyQUANT Direct Cell Proliferation Assay. Histograms depict mean ± SEM for a minimum of 4 independent experiments.

**Figure 3 toxics-12-00472-f003:**
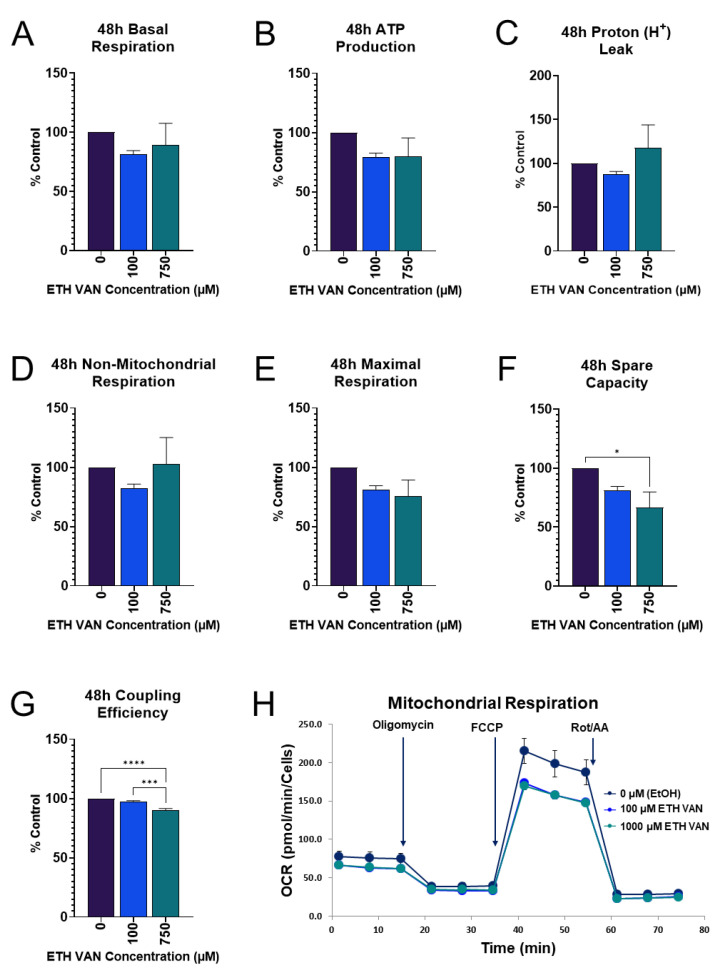
**Effects of ethyl vanillin on mitochondrial function after 48 h.** No significant changes in basal respiration, ATP production, proton leak, Maximal respiration and non-mitochondrial respiration were observed after 48 h (**A**–**E**). Spare capacity and coupling efficiency (**F**,**G**) were both significantly decreased by ETH VAN. Panel (**H**) depicts Seahorse cell mito stress test on cells exposed for 48 h to ETH VAN. Statistical differences denoted by an asterisk (* *p* < 0.05, *** *p* < 0.001, **** *p* < 0.0005). Values represent mean ± SEM for ≥4 independent experiments.

**Figure 4 toxics-12-00472-f004:**
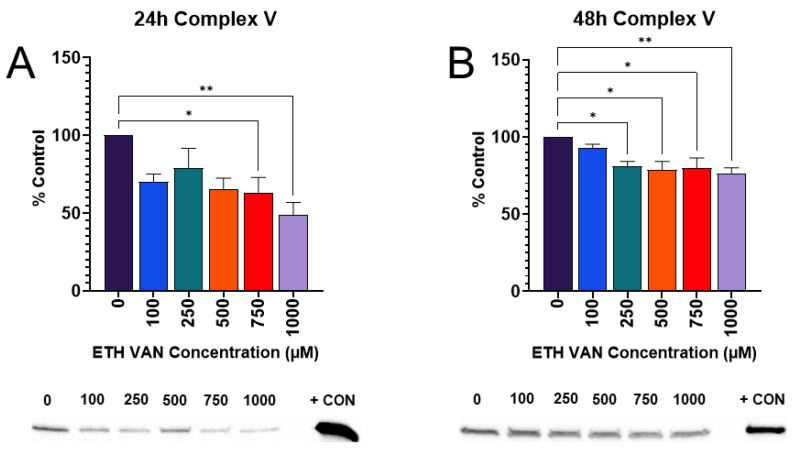
**Ethyl vanillin effects on mitochondrial OXPHOS complex V (ATP Synthase).** ETH VAN exposure significantly reduced ATP synthase expression within 24 h (**A**) and 48 h (**B**). Expression was normalized based on total protein. An asterisk denotes statistical differences (* *p* < 0.05, ** *p* < 0.01). Bar graphs represent mean ± SEM for ≥4 different experiments.

**Figure 5 toxics-12-00472-f005:**
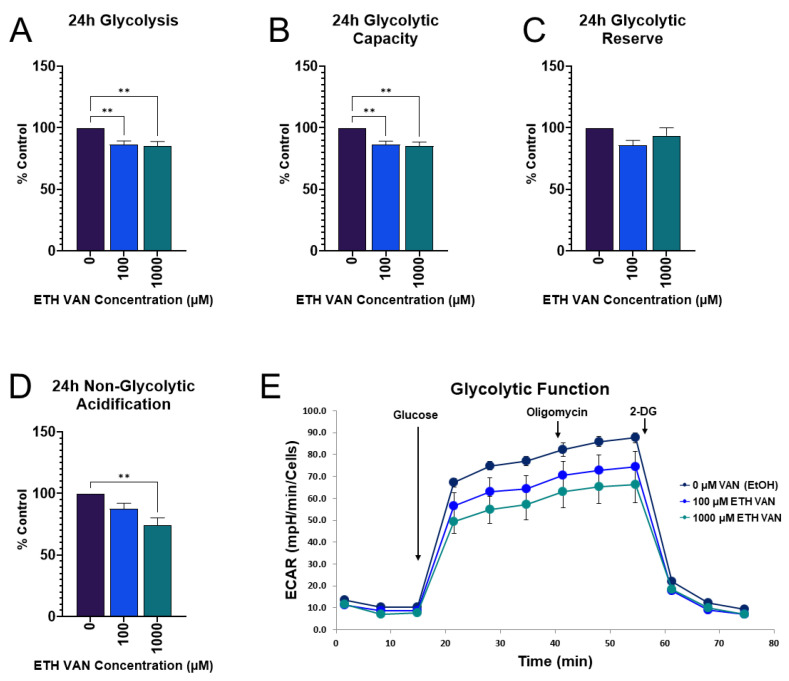
**Effects of ethyl vanillin on glycolytic function after 24 h.** Data expressed as mean ± SEM for a ≥4 separate experiments. ETH VAN decreased glycolysis, glycolytic capacity, and non-glycolytic acidification (**A**,**B**,**D**). Glycolytic reserve was not statistically different (**C**). Representative profile of glycolysis stress test (**E**). Normalization of cell number was conducted as described above. Statistical differences are indicated by an asterisk (** *p* < 0.01).

**Figure 6 toxics-12-00472-f006:**
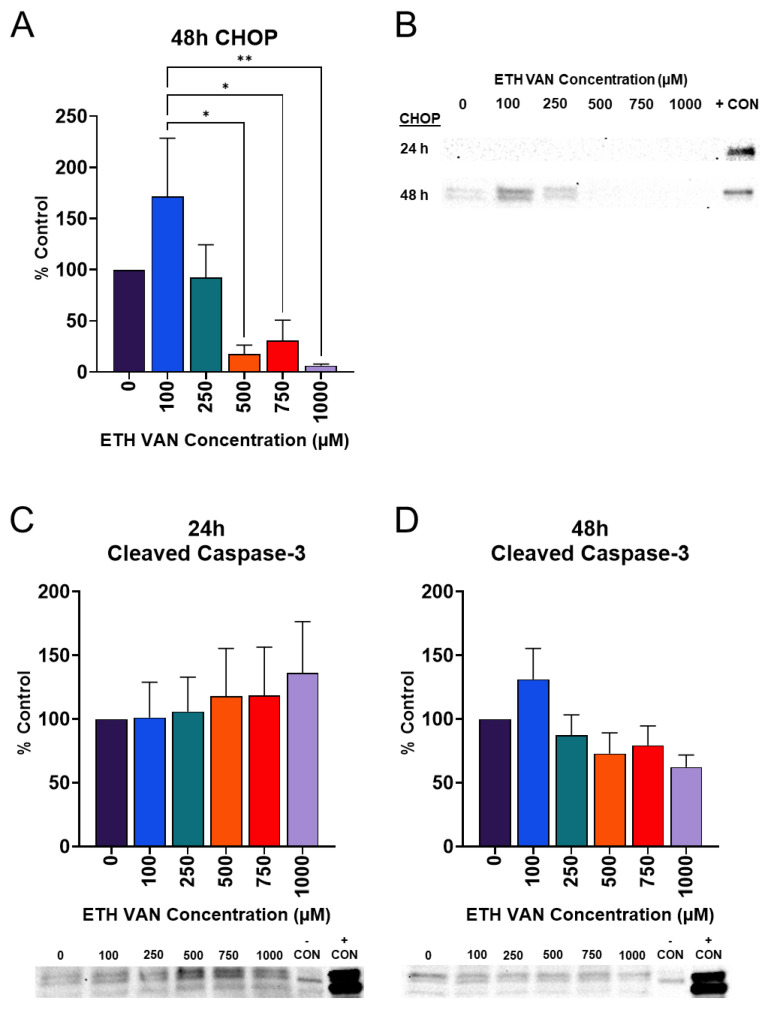
**Effects of ethyl vanillin on endoplasmic reticulum and apoptosis and stress markers.** CHOP expression was only observed after 48 h exposure to ETH VAN, and subsequently lost at higher concentrations (**A**,**B**). Cleaved caspase-3 expression showed an increasing trend (*p* > 0.05) (**C**). Cleaved caspase-3 expression (*p* > 0.05) showed a decreasing trend (**D**). C2C12 cells plus thapsigargin were the positive control for CHOP. HeLa cells +/− staurosporine were used as controls for cleaved caspase-3. Differences are indicated by asterisk(s) (* *p* < 0.05, ** *p* < 0.01). Values depict mean ± SEM for a ≥4 separate experiments.

**Figure 7 toxics-12-00472-f007:**
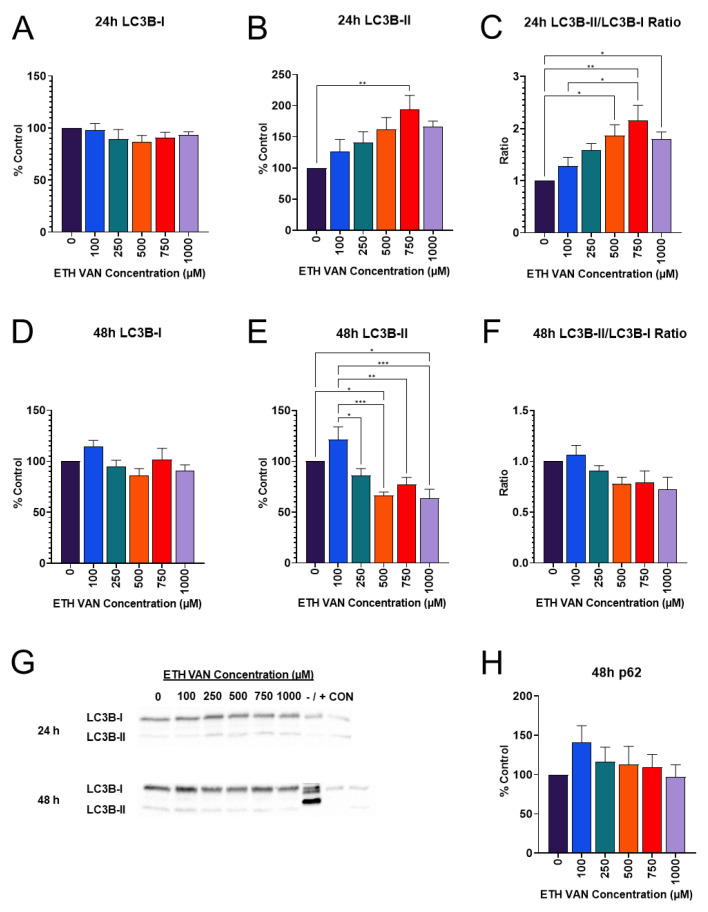
**Effects of ethyl vanillin on autophagy marker LC3B.** LC3B-I expression was unchanged at 24 or 48 h by ETH VAN (**A**,**D**). LC3B-II increased significantly after 24 h, between control and 750 µM ETH VAN (**B**). By 48 h exposure, LC3B-II expression decreased significantly (**E**). LC3B-II/LC3B-I ratio increased (**C**), but this increase was lost at later time point (**F**). Representative gel for LC3B-I and LC3B-II (**G**). HeLa cells +/− staurosporine were markers for LC3B. After 48 h, p62 showed an initial increase, followed by decrease in expression (**H**). Results were normalized to total protein. Asterisk(s) indicates statistical differences (* *p* < 0.05, ** *p* < 0.01, *** *p* < 0.001). Values represent mean ± SEM for a ≥4 separate experiments.

**Figure 8 toxics-12-00472-f008:**
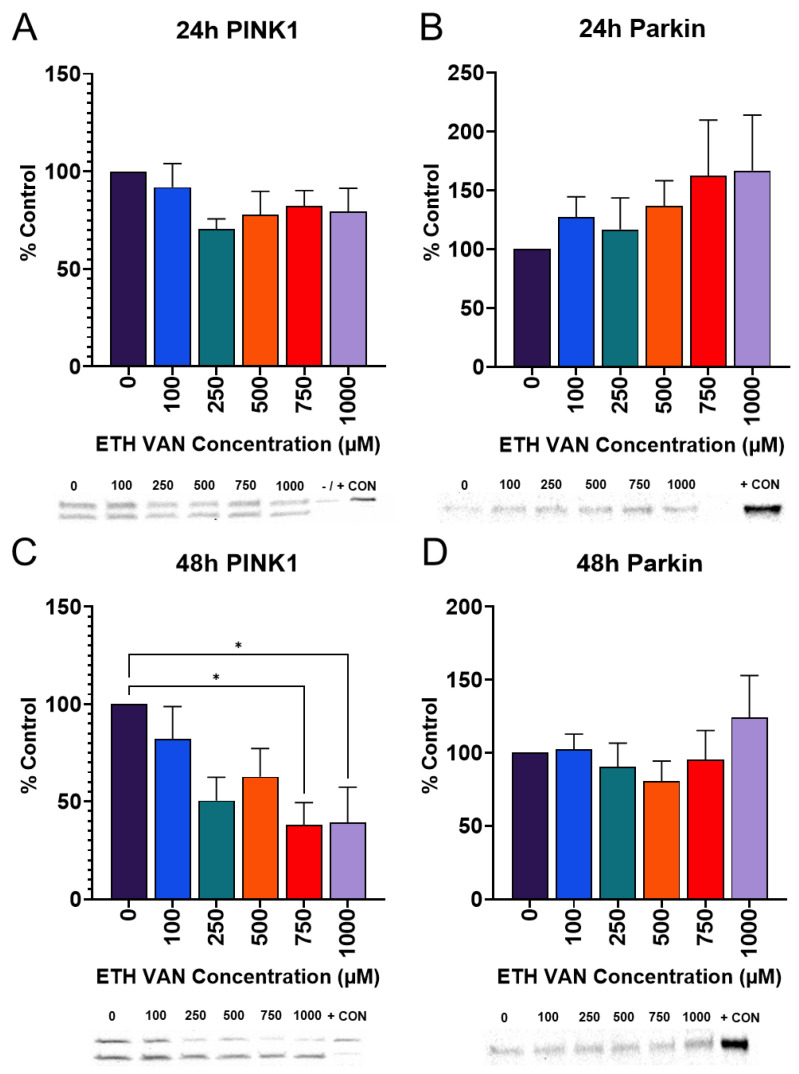
**Effects of ethyl vanillin on the mitophagy markers PINK1 and Parkin.** PINK1was not altered at 24 h (**A**). PINK1 levels declined at 48 h (**C**). Parkin was unchanged (**B**,**D**), although an increasing trend in expression was observed. HeLa cells treated with staurosporine were the PINK1 positive marker. Parkin positive control was mouse brain lysate. Results were normalized to total protein. Statistical differences denoted by an asterisk (* *p* < 0.05). Values represent mean ± SEM for ≥4 separate experiments.

**Figure 9 toxics-12-00472-f009:**
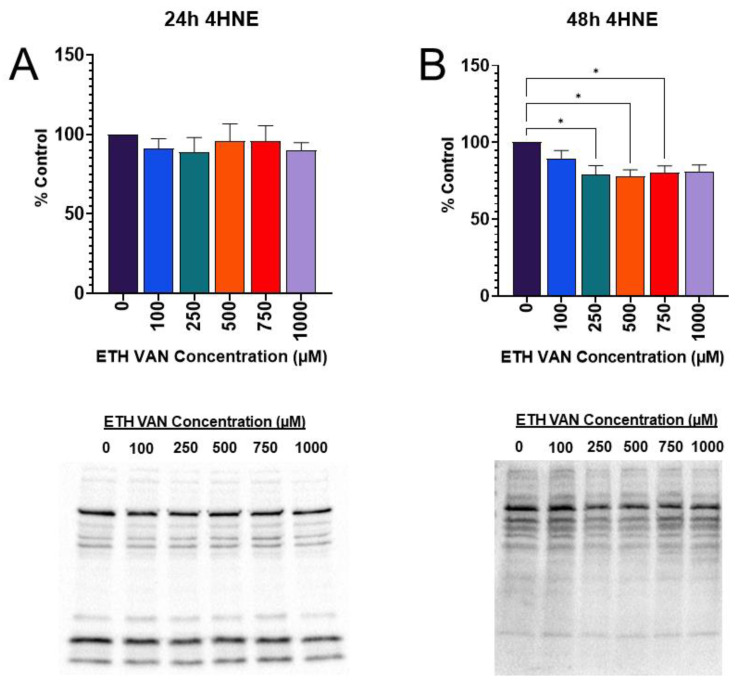
**Effects of ethyl vanillin on 4-HNE.** No significant differences in 4-HNE were observed at 24 h (**A**). After 48 h exposure, 4-HNE adduction decreased between control and 250–750 μM (**B**). Representative blots are shown below each graph. Whole-lane analysis was used for 4-HNE. Results were normalized to total protein. Statistical differences depicted by an asterisk (* *p* < 0.05). Results are mean ± SEM for ≥4 separate studies.

## Data Availability

Data will be made available upon request.

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
