# Peer review of "The Flavoring Agent Ethyl Vanillin Induces Cellular Stress Responses in HK-2 Cells"

_toxics, 2024, doi:10.3390/toxics12070472_

Round 1

Reviewer 1 Report

Comments and Suggestions for Authors

I read the manuscript entitled “The E-liquid Flavoring Ethyl Vanillin Induces Cellular Stress Responses in Renal Human Proximal Tubule (HK-2) Cells” with interest, but many doubts assail me as to how the authors can conclude that the ethyl vanillin flavor (particularly) in e-liquids is the cause of the alterations reported in the renal cells of the human proximal tubule. 

The e-liquids are composed (over 90%) of propylene glycol and vegetable glycerin; they may also contain nicotine and various flavorings (including ethyl vanillin). Studying the effects of a component of e-liquids on the cells of (potential) users of e-liquids means testing an e-liquid containing at least the main substances that compose it, plus the component (in this case ethyl vanillin), somehow mimicking the method of administration of the e-liquid (the best method would be to vaporize the e-liquid using an e-cigarette and a vaping machine that complies with ISO standards). 

In the exposure method reported for this work there is no trace of this attempt, but the substance is diluted in ethanol (why?) and added directly to the cell cultures. Therefore, the thermal degradation processes that the substance would undergo in e-cigarettes (warmed at 200-250 °C) the effect of the filtering operated by the epithelial-endothelial cellular barrier of the pulmonary alveoli or the liver metabolism (which transform part of the ethyl vanillin), before it reaches the kidneys are taken into account.

But, even if we would to overlook these aspects, the work could be modified in its aim of demonstrating the effects of ethyl vanillin contained in e-liquids on renal function, moving on to a broader demonstration of the effects of the human consumption (e-liquids, foods, beverages, cosmetic and personal care products, etc.) of ethyl vanillin on renal function. 

- In my opinion, all the introduction and conclusion should be reviewed with these consideration in mind.

But I still have a crucial doubt about the interpretation of the results reported in the manuscript. While accepting that ethyl vanillin is soluble only in ethanol or similar compounds (and this is not the case), not a single vehicle control (Ethanol alone) has been reported to exclude that the effects observed are not due (at least in part) to this substance. Ethanol has well-known toxic effects on cultured cells. Furthermore, some of the effects reported in the manuscript, and attributed to ethyl vanillin, are reported in the literature as effects attributable to ethanol. Just as an example ethanol is often used to induce ROS production on cells in vitro.

As example I suggest to take a look at some papers reporting effects of ethanol on different kind of cells (doi:10.1080/09168451.2014.921561.Epub ; 10.1073/pnas.1401853111 ; 10.4161/auto.21376)

How can the authors claim that these effects are attributable to ethyl vanillin, and they are not related to the vehicle (ethanol)?

- In my opinion all the experiments need to be repeated with adequate vehicle controls.

Author Response

Response to Review #1

We thank the reviewer for the time spent on our review. The authors of well aware of the cellular effects of ethanol and the necessity for proper vehicle controls. We have addressed the reviewer’s comments below.

COMMENT: Why was ethanol  used as the vehicle instead of water?

RESPONSE: Ethyl-vanillin is not sufficiently soluble in water to be used as a vehicle. Normally, we had 1% solvent to media. Ethyl-vanillin solubility  in water is 1 gram/100 ml or 10 grams/Liter. The MW of ethyl-vanillin is 166. The maximum concentrated solution of ethyl-vanillin would be 60 mM. The volume of water needed as a vehicle would dilute the cell media during incubation  to the point of impairing growth, nutrients and possibly pH regulation. The solubility in ethanol is 375 mg/ml  or 2.25 M. The volume of ethanol (vehicle) or ethanol with ethyl vanillin applied to plates  was a maximum of 1% of the total volume of media plus treatment in order to obtain the final concentrations of 0-1000 uM. That is the reason ethanol was used as the solvent.

COMMENT: The ethanol if denatured has other components

RESPONSE: Sect. 2.1  line 73 of the  original manuscript indicated that  ethanol was a vehicle for ethyl vanillin. Koptec was the source and this company only sells USP grade ethanol. We have added to the revise manuscript that the ethanol is USP grade 200 proof. The ethanol was not denatured, it was USP quality.

COMMENT: Ethanol vehicle treated groups were not included in the study.

RESPONSE: Reviewer #1 concluded that  the authors had not included an EtOH vehicle control run and this was not the case. All studies were conducted with a vehicle control which was run simultaneously in all plates and was an equal volume of EtOH when compared to the ethyl-vanillin group. All  experiments were conducted with a simultaneously run ethanol vehicle control. Sect. 2.2 line 87 in the original manuscript stated that  The vehicle group received an equal volume of EtOH.” We have added to the manuscript that the total volume of ethanol did not exceed 1% of the final volume per well to further clarify the amount of ethanol used in each well either as vehicle control or solvent for ethyl-vanillin. All treatment groups were compared to the ethanol treated group as the control group.

COMMENT: The effects of propylene glycol and vegetable glycerin should be examined.

RESPONSE: The authors agree that cellular effects of these components is worth exploring. Several studies are being conducted by other investigators. However, this is beyond the scope of this manuscript.

COMMENT: Why examine renal effects?

RESPONSE: Lines 55-62 discuss the publications reporting renal changes with vaping of nicotine containing or nicotine free products. These reports are clinical as well as in rodents. The clinical reports are of renal impairment associated with vaping. The clinical reports provide a rationale for examining extrapulmonary effects specifically to the kidney.

Reviewer 2 Report

Comments and Suggestions for Authors

The authors examine effects of "acute" exposure of plated human proximal tubule (HK-2) cells to ethyl vanillin followed by examination of cell viability, mitochondrial activity, etc. At times, it seems that the authors stretch the imagination to justify or explain a result, as if toxicity is expected. While flavorants with aldehyde groups may possess quite a bit of chemical irritant activity or worse, possess possible carcinogenic activity when inhaled (cinnamaldehyde, another flavorant, for example), it should be no surprise that the assays did not indicate significant toxicity, given the fact that the molecule also has a phenolic functional group, and as a consequence, its protective antioxidant activity has also been studied.

Nevertheless, although the authors discussed the flavorant's use in electronic cigarette liquids in the introduction, they did not study the effects using pulmonary cells, but rather considered renal effects. The authors need to justify why they used the flavorant rather than the carboxylic acid product of the aldehyde group, which is easily oxidized, or one of the glucuronides to which ethyl vanillin is metabolized, the form to which renal cells would most likely be exposed.

I have a few specific comments:

1. Line 72: Was 95% (distilled) ethanol used, or 100% but denatured ethanol, used which of course would contain other toxic substances?

2. Line 107: The authors are correct that the MTT assay depends on oxidoreductases catalyzing reaction of a dye. The activity is NADH dependent, indicating intact electron transport system in the mitochondrion.

3. Line 187: The authors in most cases do not mention the number of experiments used to produce each data point. In several graphs, SEM bars make the data look very tight, precise, considering biological assays. Using standard error of the mean (standard deviation divided by the square root of the number of experiments) makes error bars tighter. Why not use standard deviation instead in all graphs, to give the reader a more visible representation of how much variability existed?

4. Lines 195-196, 384-385: The way that statements in these lines makes the meaning sound like a time course (initially...subsequently). All data represents assay results at the same end time. However, Cell viability assays determine the proportion of cells that were viable at the beginning of a procedure that are still viable at a later time, or after a given perturbation. One cannot have a greater percentage of viable cells than 100% (which would imply that no cells were lost due to cell toxicity or apoptosis). What is actually shown in Figure 1A is an increase in oxidoreductase activity likely due to an increase in cell NADH concentration., not an increase in the number of viable cells greater than 100%. Perhaps the antioxidant activity of the phenolic group of ethyl vanillin acted to reduce any oxidative challenge to the mitochondria (and the cell as a whole), thus decreasing the oxidation of NADH/NADPH and increasing the availability within the cells. After 48 hours, perhaps the antioxidant capacity of the ethyl vanillin was depleted at 48 hours, and toxicity at some level impacted the mitochondrial electron transport activity, as indicated by the higher concentrations of ethyl vanillin at 24 hours. The authors should reword their statement on increased mitochondrial viability. "Activity" in line 384 is a good choice of wording. "Viability" again needs to be reworded.

5. Figure 1 caption lines 204-205: Mitochondrial health is a better way of expressing the mitochondrial effects than viability (comment 4). Still, mitochondrial health was not under examination either. Available reducing power in the form of NADH is what was being demonstrated.

6. Figure 3H: The figure symbol code shows three data point sequences. The 100 micromolar ethyl vanillin and the 1000 micromolar ethyl vanillin data points are almost identical results. It takes magnification to even see that there are two lines. Perhaps a change of the dark line to a more different color such as red would help.

7. Lines 407-413: See comment 4 and my early comments on protective antioxidant activity of ethyl vanillin, which is the topic of several published studies. Maybe taking that into consideration will aid in understanding "interesting results".

8. Lines 417-423: maybe ethyl vanillin has been metabolized or the carbonyl group oxidized to less toxic carboxyl group after 48 hours.

Author Response

Response to Reviewer #2

We thank the reviewer for their insightful evaluation of our manuscript.

COMMENT: Why examine renal effects?

RESPONSE: Lines 55-62 discuss the publications reporting renal changes with vaping of nicotine containing or nicotine free products. These reports are clinical as well as in rodents. The clinical reports are of renal impairment associated with vaping. The clinical reports provide a rationale for examining effects specifically to the kidney.

COMMENT: Was 95% or 100% ethanol or denatured alcohol used?

RESPONSE: Sect. 2.1  line 73 of the  original manuscript indicated that  ethanol was a vehicle for ethyl vanillin. Koptec was the source and this company only sells USP grade ethanol. We have added to the revised manuscript that the ethanol is USP grade 200 proof. The ethanol was not denatured, it was USP quality.

COMMENT: MTT Assay  depends on oxidoreductases. The activity is NADH dependent.

RESPONSE: Lines 112-113 listed NAD(P)H which include NADH and NADPH  as suggested in the following reference (Blacker TS, Duchen MR. Investigating mitochondrial redox state using NADH and NADPH autofluorescence. Free Radic Biol Med. 2016 Nov;100:53-65. doi: .1016/j.freeradbiomed.2016.08.010. Epub 2016 Aug 9. PMID: 27519271; PMCID: PMC5145803.

COMMENT: The authors in most cases do not mention the number of experiments used to produce each data point.

RESPONSE: Lines 90-91 state 4 independent experiments with different passages. Lines 196-197 states in the statistical component that 4 independent experiments were conducted. Figure legends 1-9 all stated 4 independent experiments with different cell passages.

COMMENT: Why use SEM instead of Standard Deviation as the data appears tight.

RESPONSE: First, the journal does not have a requirement regarding the expression of data as SEM or SD.  It was our preference to use SEM. Expressing data as Mean ± SEM is generated after the statistical analysis. The statistical tests used raw data and the differences between groups are not biased as to whether we express as Mean ± SEM  or Mean ± S.D.

COMMENT: Reword the MTT results to remove viability and change to activity.

RESPONSE: It was not the authors intent to infer that the number of cells was increasing within a 24 h period. We agree with the reviewer to replace the word “viability” with activity and this has been highlighted on lines 201 and 204.

COMMENT: Figure 3H has 2 different colors  for the 100 and 1000 uM ETH VAN.

RESPONSE:The colors in the Mitochondrial assay were matched  in colors to all the bar graphs in the panels for ease of viewing.

COMMENT:  CHOP and apoptosis related to antioxidant activity of ETH VAN.

RESPONSE: We agree with the reviewer. Lines 455-456 mention that the results may be similar to reported antioxidant activity which was cited earlier in the manuscript.

Reviewer 3 Report

Comments and Suggestions for Authors

I would like to congratulate the authors for putting the manuscript together. I have following enquiries/suggestions regarding the manuscript. 

1. In abstract, line 14, write the full forms of mitochondrial functions you analyzed. short forms are ambiguous not accurate way to present. 

2. line 22, when "LC3B-II to LC3B-I showed a decreasing trend" just write a sentence describing what does that mean in your studies?

3. Can authors explain "what is the concentration of ETH VAN in human blood/body and what will be its concentration in Kidney". How do authors justify the use of 1mM of ETH VAN for studies?

4. line 72, Can authors please describe how much alcohol is needed to dissolve ETH VAN? also alcohol controls are missing in many experiments, can authors explain?

4. line 79, Can authors confirm the amount of antibiotic use in there studies? in current form antibiotic concentration seems too high. 

5. authors please format the cell numbers properly across the manuscript. 

6. line 93, How did authors collected the cells? by scarping? or trypsinization? in both techniques we can observe cell death. how did authors addressed the issue?

7. for MTT assay, 35K cells in 96 well mode seems too high and there wont be any room of cells to attach and grow. in addition authors carried out experiment for 4 days. can authors justify there protocol?

8. line 111, authors are advised to use scientific writing. for example, cells were plated or cultured in 6 well plate..etc 

9. In MTT results, Authors use term "MTT assay for mitochondrial viability". As for as MTT is concern it measures mitochondrial function not the viability. can authors confirm there claims ?

10. in seahorse data, in X axis what is meant by %control? 

11. Do authors run all the complex on a single blot? if yes please provide full blot image. Also authors are suggested to provide house keeping blot. 

12. can authors explain what is +CON means in the figures? please discuss in the figure legends.

13. how did authors measured the ATG7 and p62 expression? if by westerns, please include the blots along with house keeping blots. 

14. Controls are missing for most of the blots. please introduce wherever necessary. 

Comments on the Quality of English Language

Moderate editing of English language required

Author Response

Responses to Reviewer #3

The authors would like to thank the reviewer for their critique of our manuscript. Our replies to your specific comments are below.

COMMENT: Abstract line 14 abbreviations

RESPONSE: The journal has a 250 word limit for the abstract. The current abstract has a word count of 249 and the abbreviation of activities was written in order to fit the journal requirements.

COMMENT: LC3BII/LC3BI ratio importance.

RESPONSE: Lines 20-23 have been revised to conclude that the increased LC3B-II expression and ratio of LC3B-II to LC3B-I suggest induction of autophagy 24 h exposure which disappeared at 49 h due to the decline in LC3BII.  There is a 250 word limit and the abstract is at 250 words.

COMMENT: What is the concentration  of ETH VAN in blood and   how do you justify a 1mM concentration?

RESPONSE: There is a paucity of data on levels following vaping. Part of the issue is the lack of regulation which allows for a   large range of flavoring concentrations in various products on the market. The other issue is that many individuals make their own flavoring combinations which have been reported to be in the mM range.  Lines 503-514 in the revised manuscript discuss the concentrations in various products.  References [9, 38 and 39]  reported levels that approached 114 mM for ethyl vanillin in the commercial or do-it-yourself products. The level of 1 mM  used in our HK-2 cells was much lower than what has been reported in these products.  

COMMENT: How much ethanol  was used as the vehicle?

RESPONSE:  We used ethanol as the vehicle as Ethyl-vanillin is not sufficiently soluble in water to be used as a vehicle . Normally, we use a maximum of 1%  alcohol solvent in the total volume of  media. The MW of ethyl-vanillin is 166. The solubility in ethanol is 375 mg/ml  or 2.25 M. The volume of ethanol (vehicle) or ethanol with ethyl vanillin applied to plates  was a maximum of 1% of the total volume of media plus treatment in order to obtain the final concentrations of 0-1000 uM. Line 73 states it was 200 proof USP specification ethyl alcohol. Lines 87-91 indicate the volume of ethanol in media.

COMMENT: Alcohol controls were missing,

RESPONSE: Line 87 states the vehicle group received an equal volume of EtOH. Lines 87-89 state the total volume did not exceed 1% of the final volume in each well. For example, 20 ul ethanol or ethanol+ETH VAN plius1980 ul media for a total volume of 2000 ul.

COMMENT: Authors should confirm amount of antibiotic

RESPONSE: The amount listed in the original manuscript was the amount 2.5 ml added to 500 ml media, apologies for the confusion. On lines 81-82 we have revised to designate it is a 0.5% solution with a level of 50 ‘Units/ml of pen-strep. We use a lower antibiotic level to prevent issues with the HK-2 cells. Other nonrenal cell lines such as DMS 53 and DMS 114  are incubated with 1% or 100 Units/ml pen-strep in the media.

COMMENT: The number of cells in a 96 well plate was 35,000

RESPONSE: The cell number was incorrect.  The number of cells in the 96 well plate for MTT was 1 x 105 HK -2 cells  which was 200 μl of a 50,000 HK-2 cells/ml solution.

COMMENT: The cells were run for 4 days.

RESPONSE: Cells are initially plated in 6 or 96 well plates and then equilibrated for 48 h to improve adherence to the plate. The media is removed and replaced with fresh media after 48 h prior to treatment with ethanol vehicle or ETH VAN for 24 h or 48 h (which would be 3 or 4 days, respectively). The vehicle and ETH-VAN wells for each experiment come from the same T75 flask  which is 1 passage and are on the same plate and run simultaneously.  Any variability is accounted fo since the vehicle and treated wells are handled at the same time.

COMMENT: use scientific term of culls were plated or cultured in 6 well plates

RESPONSE: Line 118 was modified and now states Cells were cultured in 6 well plates.

COMMENT: MTT  assay  is a measure of mitochondrial function

RESPONSE: We have modified the subtitle for 3.1 to  MTT Activity (lines  202-205) . Figure 1  legend revised to mitochondrial health (line 217).

COMMENT: Seahorse Data what is meant by % of control?

RESPONSE: The raw value averages after normalizing to cell numbers is shown in panel H of Figures 2 and 3 as well as Panel E of Figure 5. The ETH VAN treated samples values were compared to ethanol vehicle treated wells and expressed as % of the ethanol vehicle treated groups.

COMMENT: What are the +CON in the figures on some of the blots?

RESPONSE: Figure legends for Figures 6, 7, and 8 have been revised to include the  +CON  treated cells in the blots.

COMMENT: entire gels should be provided

RESPONSE: The gel images are included in the supplemental files as requested by the journal. All gels were normalized to total protein scan of the respective lanes. This was done because some cytoxic substances can alter housekeeping proteins such as GAPDH and actin.

COMMENT: Controls missing from blots

RESPONSE: All blots were run from samples collected at the same time from a plate that had 0-1000 uM ETH VAN. The 0 uM ETH VAN contained an equal volume of ethanol and was the control for each protein of interest.  

Round 2

Reviewer 1 Report

Comments and Suggestions for Authors

I would like to thank the authors for their replies, but reading their results, some other doubts arose about the methods and results. I list these doubts as follows: 

1. MTT assay, material and methods section: is 1x104 the TOTAL amount of cells seeded for each plate? Or is this amount the total for each well?

2. During the MTT assay, why were the plates incubated at room temperature instead of 37°C with 5% CO2? I think that incubating the cells for 4 hours in absence of controlled conditions could affect the final result of the viability assay. 

3. Western Blot analysis: Which protein did the authors consider as internal control?

4. Western Blot analysis, CHOP expression at 48h: The figure does not show any band at 500, 750 and 1000 uM. Why do the histogram columns related to these concentrations show some results?

5. Western Blot analysis, cleaved caspase-3: looking at the supplementary materials, and specifically at the pictures used for the figures, how did the authors normalize the obtained results? In addition, check the 24h graph for errors. 

6. Western Blot analysis: Why did the authors not show the bands figure for p62 protein?

7. Western Blot analysis, Pink1 /Parkin: Why did the authors normalize results on protein concentration? Where is the internal control on the membrane? (supplementary materials). 

Moreover, I would appreciate the title and conclusion being adjusted because the setup of the exposure to ETH VAN is not conceived to test the effects of this chemical flavor in e-liquids (and used with an electronic cigarette), but it is configured only to test (in a generic way) the effects of this chemical flavor on Renal Human Proximal Tube (HK-2) cells. Therefore, the authors' observations may be relevant for the use of this substance in general (food, drinks, cosmetic products, and even e-cigarettes). I would like this aspect of the research to be well highlighted in the scope of the work and the conclusions, otherwise, there is a risk of sending incorrect information to the scientific community and regulatory bodies.

I would add some comments on ethanol that was used at 1% concentration (it is generally suggested to use it around 0.1% concentration). But, if you say that the “0” column in your graphs represents the cells treated with ethanol at the same concentration of cells (I would expect five different controls for five different treatment concentration, unless you say that what you show is the highest concentration of ethanol you used), could you add to your graphs an untreated control, with no treatments and no ethanol? Or you could add a comparison between untreated cells and ethanol only treated cells to the supplementary material?

Author Response

Responses to Second Review by Reviewer #1

  • Line 106 has been revised to clearly state that 10,000 cells are in each well. The changes are in green highlight. It is not evident why the assumes the number of cells were used for the entire plate since the number of wells one uses per plate is dependent on the number of treatments in an experiment.

  • All groups were under the same conditions for the MTT assay. The 96 well plates contained  all treatments 0 (ethanol), 100, 250, 500, 750 and 1000 uM 0f ethyl vanillin  and were all given the same amount of MTT and were incubated for 4 h  at 72 degrees (room temperature). All groups were in the same conditions so the differences are mediated by the treatments.

  • For western blots,  the  protein densitometry in the entire  lane was  used to normalize the protein of interest. Total protein was visualized with Memcode  staining and calculated for each lane.  For example, the total protein for each 100 uM treated ETH VAN lane was determined and then used  as a ratio to the densitometry for a protein of interest such as   The vehicle control of ethanol denoted 0 uM values were set at 100 % and the values were then compared between the treatment groups of 100-1000 uM. The Memcodes for each gel are included with the westerns in the supplementary material. The reason for not using GAPDH or β-actin is that there are publications which show these traditional housekeeping proteins are altered by toxic substances or disease states. These  are just 3 publications reporting alterations in housekeeping proteins. The Dietze was published by a very respected toxicologist, Syd Nelson  which showed acetaminophen toxicity impacts GAPDH.

 (Zhang B, Wu X, Liu J, Song L, Song Q, Wang L, Yuan D, Wu Z. β-Actin: Not a Suitable Internal Control of Hepatic Fibrosis Caused by Schistosoma japonicum. Front Microbiol. 2019 Jan 31;10:66. doi: 10.3389/fmicb.2019.00066) and

(Comajoan P, Gubern C, Huguet G, Serena J, Kádár E, Castellanos M. Evaluation of common housekeeping proteins under ischemic conditions and/or rt-PA treatment in bEnd.3 cells. J Proteomics. 2018 Jul 30;184:10-15. doi: 10.1016/j.jprot.2018.06.011).

Dietze EC, Schäfer A, Omichinski JG, Nelson SD. Inactivation of glyceraldehyde-3-phosphate dehydrogenase by a reactive metabolite of acetaminophen and mass spectral characterization of an arylated active site peptide. Chem Res Toxicol. 1997 Oct;10(10):1097-103. doi: 10.1021/tx970090u.

  • If you magnify the gel images, there is a weak band for CHOP.

  • The fuzzy %control on the y-axis of Figure 6 panel C has been revised.

  • The bands for p62 were not included as they were not different between groups. They are in the supplementary material.

  • Pink1 and Parkin were normalized to total protein densitometry for their respective lanes. Protein was evaluated for total lane densitometry same as described in item 3. It is not normalized to protein concentration as the same volume and amount of protein was loaded in each lane.

Par 1 comments. The title states Ethyl Vanillin is a flavoring agent and does not infer we are examining e-liquids. The title states ethyl vanillin induces cellular changes in HK2 cells. Titles of other publications  have used E-cigarette flavorings and it is not clear why the reviewer opposes our title. The reviewer should check the following publications as examples (Hickman E, Herrera CA, Jaspers I. Common E-Cigarette Flavoring Chemicals Impair Neutrophil Phagocytosis and Oxidative Burst. Chem Res Toxicol. 2019 Jun 17;32(6):982-985. doi: 10.1021/acs.chemrestox.9b00171. 

Martin A, Tempra C, Yu Y, Liekkinen J, Thakker R, Lee H, de Santos Moreno B, Vattulainen I, Rossios C, Javanainen M, Bernardino de la Serna J. Exposure to Aldehyde Cherry e-Liquid Flavoring and Its Vaping Byproduct Disrupt Pulmonary Surfactant Biophysical Function. Environ Sci Technol. 2024 Jan 23;58(3):1495-1508. doi: 10.1021/acs.est.3c07874. 

Rickard BP, Ho H, Tiley JB, Jaspers I, Brouwer KLR. E-Cigarette Flavoring Chemicals Induce Cytotoxicity in HepG2 Cells. ACS Omega. 2021 Mar 2;6(10):6708-6713. doi: 10.1021/acsomega.0c05639. 

Lines 519-528 are the authors conclusions  and this section only mentions the effects of ETH VAN on HK-2 cells and does not focus on e-liquids.

Par 2 of comments. There is more than 1 way to treat cells with varying concentrations of a test compound. Perhaps the reviewer was thinking we make 1 solution that is 100x for the 1000 uM ethyl vanillin and then add diminishing volumes to the wells to generate the final concentrations of 250,    500 and 750 uM. However, this is not how we treated the cells.

In our study, all wells were given the same volume of ethanol which  was 1% of the final volume of media and 0-1000 uM ethyl vanillin. The various concentrations of ethyl vanillin were prepared so that all concentrations  were added to the wells at the same volume. We are not adding 5 different volumes of ethanol to make the varying ethyl vanillin final concentrations.  We prepare 5 dilutions of ethyl-vanillin in ethanol  that when added provide the final concentration between 0-1000 uM. These dilutions are 100x or (10, 25, 50, 75 or 100 mM) and when added as 1% of final volume yield 0, 250, 750 or 1000 uM. For example,  the 6 well total volume was 2000 ul then 1980 ul of media with cells was in the well and 20 ul of ethanol was added yielding a final concentration of 0-1000 uM. Lines 108-109 have been revised to state the treated and vehicle group received the same volume.

Reviewer 2 Report

Comments and Suggestions for Authors

I thank the authors for addressing comments. The NADH/NADPH response appears to be more of a sidestep to the question, but based on the citation, I will let it pass. It was important to state "USP" grade ethanol since 200 proof may also contain benzene, methanol, acetone, and/or alkanes and could contribute to toxicity. Thanks for adding the specification of USP grade.

Author Response

We would like to thank the reviewer. The USP grade ethanol has been used in our lab as the vehicle as we are aware of the other chemicals in denatured alcohol. Thank you.

Reviewer 3 Report

Comments and Suggestions for Authors

all the suggested corrections were made in the manuscript. 

Author Response

Reviewer #3 voted to approve for publication. We thank the reviewer for their comments.  

Round 3

Reviewer 1 Report

Comments and Suggestions for Authors

I would like to thank the authors for their replies. Below are some comments.

·       Thank you to have revised the text. I can understand that the authors could find my comment superficial or not useful, since they think I’m assuming something without reason, but if I read this sentence “A total of 1 x 10HK -2 cells were plated on 96-well culture plates (Fisher Scientific, Item No. FB012931) and allowed to grow for 48 h”, it clearly means “a total in a plate”. I think it was simply a typo, I think there is no problem behind that, especially in correcting it.

·       I totally agree with the authors: there are some cases in which it is not possible to use a housekeeping protein as internal control, due to the specific conditions used during the experiment. But I think that they are not in that condition. I think that, if possible, the use of an internal control could represent the most crucial part of a western blot analysis. First, it gives clues regarding the conduction of the experiment, confirming the quality of the technique; second, the normalization following an internal control loading allow you to handle the membrane the least possible, enhancing the chance to have a clear membrane to be acquired. Moreover, there are several proteins that could be used as internal control, so you can easily choose depending on your specific situation. 

Anyway, although it is considered a good technique, I suggest to use the total protein normalization (TPN) approach only if really needed, or specifically required by the journal (and in my opinion, this is not the case).

·       Of course I did magnify the image, and I can ensure the authors that the image they have proposed in the paper does not show any band at 500, 750 and 1000 uM. If I look at the supplementary materials, I can see some interesting details: the authors write about CHOP (26 kda); if I follow the kda levels, starting from the first figure on the left, I can see that the strip related to the one reported in the paper (image in the center) is referred to 40 kda, so it is not CHOP. The strip related to 26 kda shows for sure bands, but the authors showed something different in the paper. The bands go slightly down moving to the figure on the right. At that point, the strip referred to 26 kda is related to the one proposed in the paper, and again, 500, 750 and 1000 concentrations show no bands (also 250 is technically empty). Finally, the figure in the center, the one used for the paper, shows a strip on the top (at 35 kda) and one at the bottom (around 20 kda) of the membrane. The one at the top, 35 kda, the one chosen and reported in the paper, shows no bands at 500, 750 and 1000. The one at the bottom, around 20 kda, shows bands. In conclusion, if the CHOP strip is the one below the one the authors showed, please change the figure in the paper. 

·       Thanks for revising figure 6, but please, check the figure 6C for a typo on Y axis

·       I would clarify that I oppose the title (and the introduction and the conclusion) because it is misleading. When you write “The E-liquid Flavoring Ethyl Vanillin Induces Cellular Stress Responses in Renal Human Proximal Tubule (HK-2) Cells”, you suggest to readers that e-liquids contain a substance that induces cellular stress responses in HK-2 cells. It is correct, but it is not accurate. It would be more accurate to write “The Flavoring Ethyl Vanillin Induces Cellular Stress Responses in Renal Human Proximal Tubule (HK-2) Cells”. In real life, as I have already written previously, this substance is used in a vast range of products, and the way in which you test the substance (which has nothing to do with e-cigarettes) is more like direct contact than kidney cells may have with ethyl vanillin taken through food or drink or, better yet, through the use of cosmetic products. Why did you decide to specify in the title that ETH VAN is a flavor of e-liquids, and not that it is a flavoring agent for other products? ETH VAN is much more used in foods than in e-liquids, and you should discuss these informations in your introduction as well as in discussion. Why not just talk about the flavoring agent ETH VAN?

You can find a wide range of studies on ETH VAN (https://ntrl.ntis.gov/NTRL/dashboard/searchResults.xhtml). Your study would add an interesting piece to the puzzle, useful for determining the safety of using this flavor in human consumption, rather than simply in vaping.

The misleading literature referring to “electronic cigarettes”, “e-liquids” or “ENDS” is accumulating in scientific literature and risks jeopardizing a product which, despite all its limitations, could represent a valid alternative to very harmful cigarette smoking. In my opinion it is not correct to keep the word "e-liquid" in the title and to conclude by the sentence “the e-liquid flavoring ETH VAN can induce energy pathway dysfunction and cellular stress responses in a renal model” or “we need more studies to assess the safety of using this flavoring in ENDS products”. If previous papers (different from Toxicology) have had the consent from other reviewers, less sensitive to the topic, of maintaining a misleading title and conclusions, it does not mean that we all have to adapt to incorrect choices of the past and of other journals. 

Moreover, reading at the papers cited by authors, the one from Hickman et al. is even more incorrect than their manuscript, given that the methods of exposure of neutrophils to the chemical flavor resemble an intravenous administration of the substance more than an exposure through e-liquids (vaporized by an e-cigarette). The paper from Martin et al. is really different from the manuscript we are discussing here, and they exposed cells (pulmonary cells) in a way that mimics the exposure to aerosol. Finally, in the manuscript from Rickard et al., there is another wrong exposure method, even if authors tried to resemble the exposure with PG/VG (used as basis of e-liquids). But it still sends a distorted message, because it should be reported to regulatory bodies that these flavors, currently approved for regular human consumption in foods, drinks, e-liquids and cosmetic products, can have harmful effects on the liver (or kidneys), proposing an extensive review of the literature to the Flavor and Extract Manufacturers Association (FEMA) of the United States. 

All your manuscript is focused on discussing about e-liquid and e-cigarette, but all the experimental part has nothing to do with e-liquids and ENDS. You should adjust the abstract, the introduction, and the conclusion according to your experimental part. In the discussion you can certainly refer to the fact that this flavor is used in e-liquids and vaping, too.

Can you demonstrate the same effects by exposing KH-2 cells to e-liquids containing ethyl vanillin “vaporized” (and bubbled through culture media or buffer conditioned with vapor) at a rational concentration? If it is not the case, I would recommend to discuss about the effects of the food chemical flavor ethyl-vanillin in a more general manner, and not as if you were suggesting that it is a chemical flavor expressly used in e-liquids and that a human being can incur the health risks you discuss just by using e-cigarettes with e-liquids containing this flavoring.

I solicit for a title and a correct and honest conclusion based on your experimental data and not on speculations that have nothing to do with science, but with the market.

Author Response

Responses to Reviewer #1 Round 3

All changes to the manuscript are in teal highlight.

Reviewer Comment: Reviewer demands title be changed.

RESPONSE: We have revised the title of the manuscript to conform to the reviewer’s demand. The title is now The Flavoring Agent Ethyl Vanillin Induces Cellular Stress Responses in Renal Human Proximal Tubule (HK-2) Cells

COMMENT: I can understand that the authors could find my comment superficial or not useful, since they think I’m assuming something without reason, but if I read this sentence “A total of 1 x 104 HK -2 cells were plated on 96-well culture plates (Fisher Scientific, Item No. FB012931) and allowed to grow for 48 h”, it clearly means “a total in a plate”. I think it was simply a typo, I think there is no problem behind that, especially in correcting it.

RESPONSE: The revised  manuscript submitted April 28th  was modified to  state on line 106 that A total of 1 x 104 HK -2 cells were plated in each well of a  96-well culture plates. This was addressed in the April 28th Revision submission. The cell number per well was highlighted in green. Perhaps the reviewer was looking at Submission 1 of our manuscript.  

COMMENT: Figure 6C has a typo on the Y-axis.

REPLY: Figure 6 panel C was revised on the Y-axis in the revision submitted April 28th. None of the authors can find an error on the Y-axis of Figure 6C. Perhaps the reviewer is looking at the first submission which was revised?  

COMMENT: The CHOP protein

RESPONSE: The CHOP protein was at 26 kDa. The verification of the protein was done with our positive control which was included in Figure 6. The positive control was also included in all supplementary gels. We have included all 4 samples used for CHOP. The images are developed on a Chemi Doc and analyzed using Image J. The image is of higher quality than can be viewed in the figures and although the reviewer’s impression is that there is no image for the 500-1000 uM  exposures there is a band. We have redone the supplementary gel images for figure 6 and these may be acceptable to the reviewer.

COMMENT: The misleading literature referring to “electronic cigarettes”, “e-liquids” or “ENDS” is accumulating in scientific literature and risks jeopardizing a product which, despite all its limitations, could represent a valid alternative to very harmful cigarette smoking.

REPLY: The citations in our manuscript are published articles that underwent peer review. If the reviewer has issues with these articles, then this reviewer should contact the journals that published these articles. This comment suggests a bias regarding this subject.

COMMENT: The manuscript focuses on e-liquids.

REPLY:  The repurposing of agents either for changes in route of delivery (oral to inhaled)  has resulted in an increase in concerns regarding short- and long-term health effects.  The original as well as revision #1 and #2 only had the term e-liquid used 1 time in the title, abstract and conclusion. We have deleted the term e-liquid from all 3 of these sections of the manuscript.  We also stated that the authors stated in the original and subsequent revisions that a limitation of our study is the addition of the flavoring agent to HK-2 cells. We did not state we were mimicking e-liquid delivery.

COMMENT: Western blot normalization should be a housekeeping protein.  

RESPONSE: There is no standardized way to normalize western blots required by the journal Toxics or MDDPI.  Second, it is easier to use GAPDH or Beta actin as it can be added to an antibody cocktail while our total protein method verifies transfer for the entire gel and not just select Molecular weights. As mentioned in Revision #2,   the  protein densitometry in the entire  lane was  used to normalize the protein of interest. Total protein was visualized with Memcode  staining and calculated for each lane.  For example, the total protein for each 100 uM treated ETH VAN lane was determined and then used  as a ratio to the densitometry for a protein of interest such as  LC3BI. The vehicle control of ethanol denoted 0 uM values were set at 100 % and the values were then compared between the treatment groups of 100-1000 uM. The Memcodes for each gel are included with the westerns in the supplementary material. The reason for not using GAPDH or β-actin is that there are publications which show these traditional housekeeping proteins are altered by toxic substances or disease states. These  are just 3 publications reporting alterations in housekeeping proteins. The Dietze was published by a very respected toxicologist, Syd Nelson  which showed acetaminophen toxicity impacts GAPDH. Further, the article by Taylor and Posch (2014) describes the shortcoming of using a traditional housekeeping protein.

 (Zhang B, Wu X, Liu J, Song L, Song Q, Wang L, Yuan D, Wu Z. β-Actin: Not a Suitable Internal Control of Hepatic Fibrosis Caused by Schistosoma japonicum. Front Microbiol. 2019 Jan 31;10:66. doi: 10.3389/fmicb.2019.00066) and

(Comajoan P, Gubern C, Huguet G, Serena J, Kádár E, Castellanos M. Evaluation of common housekeeping proteins under ischemic conditions and/or rt-PA treatment in bEnd.3 cells. J Proteomics. 2018 Jul 30;184:10-15. doi: 10.1016/j.jprot.2018.06.011).

Dietze EC, Schäfer A, Omichinski JG, Nelson SD. Inactivation of glyceraldehyde-3-phosphate dehydrogenase by a reactive metabolite of acetaminophen and mass spectral characterization of an arylated active site peptide. Chem Res Toxicol. 1997 Oct;10(10):1097-103. doi: 10.1021/tx970090u.

Taylor SC, Posch A. The design of a quantitative western blot experiment. Biomed Res Int. 2014;2014:361590. doi: 10.1155/2014/361590. Epub 2014 Mar 16. PMID: 24738055; PMCID: PMC3971489.

COMMENT: Vaporize and bubble ethyl vanillin in media

RESPONSE: First, we deleted e-liquid from the title, abstract and conclusion as demanded by the reviewer. This changes the manuscript  to characterize cytotoxicity of ethyl vanillin. As mentioned previously, we have changed the title and deleted e-liquid from the abstract and conclusion.  The proposed methods are outside of the realm of this manuscript. We do not have instrumentation to heat flavors and then pass over cells or bubble into media.

Round 4

Reviewer 1 Report

Comments and Suggestions for Authors

I appreciated that the authors took my suggestions for the title and conclusions into account. All the revisions were resolved and discussed acceptably. Thanks